# Changes in Soil-Borne Communities of Arbuscular Mycorrhizal Fungi during Natural Regrowth of Abandoned Cattle Pastures Are Indicative of Ecosystem Restoration

**Carlos H. Rodríguez-León** [1,2]**, Clara P. Peña-Venegas** [3,]*****, Armando Sterling** [2]**, Herminton Muñoz-Ramirez** [2] **and Yeny R. Virguez-Díaz** [2]

[1] Doctoral Program in Natural Sciences and Sustainable Development, Faculty of Agricultural Sciences, Universidad de la Amazonía, Florencia 180001, Caquetá, Colombia; crodriguez@sinchi.org.co

[2] Laboratory of Phytopathology, Amazonian Scientific Research Institute Sinchi—Faculty of Basic Sciences, Universidad de la Amazonía, Florencia 180001, Caquetá, Colombia; asterling@sinchi.org.co (A.S.); hermintonr@gmail.com (H.M.-R.); ynyro-17@hotmail.com (Y.R.V.-D.)

[3] Laboratory of Microbiology, Amazonian Scientific Research Institute Sinchi, Leticia 910001, Amazonas, Colombia

***** Correspondence: cpena@sinchi.org.co; Tel.: +57-3108149907

**Abstract:** Natural restoration of ecosystems includes the restoration of plant-microbial associations; however, few studies had documented those changes in tropical ecosystems. With the aim to contribute to understand soil microbial changes in a natural regrowth succession of degraded pastures that were left for natural restoration, we studied changes in arbuscular mycorrhizal (AM) fungal communities. Arbuscular mycorrhizal fungi (AMF) establish a mutualistic symbiosis with plants, improving plant nutrition. Amplification of the small subunit rRNA with specific primers and subsequent Illumina sequencing were used to search soil-borne AM fungal communities in four successional natural regrowth stages in two landscapes (hill and mountain) with soil differences, located in the Andean-Amazonian transition. Molecular results corroborated the results obtained previously by spores-dependent approaches. More abundance and virtual taxa of AMF exist in the soil of degraded pastures and early natural regrowth stages than in old-growth or mature forest soils. Although changes in AM fungal communities occurred similarly over the natural regrowth chronosequence, differences in soil texture between landscapes was an important soil feature differentiating AM fungal community composition and richness. Changes in soil-borne AM fungal communities reflect some signals of environmental restoration that had not been described before, such as the reduction of *Glomus* dominance and the increase of *Paraglomus* representativeness in the AM fungal community during the natural regrowth chronosequence.

**Keywords:** AMF; degraded pastures; natural regrowth succession; soil texture; tropical soils

## 1. Introduction

Arbuscular mycorrhizal (AM) symbiosis is a plant-fungal association that occur in most terrestrial plants and almost in every terrestrial ecosystem on the planet, with important roles in plant nutrition and soil structure [1,2]. In natural ecosystems, plants and arbuscular mycorrhizal fungi (AMF) have evolved together, adjusting mutually for the best mutualistic association [3–5]. However, AM fungal communities are affected by changes in soil conditions [6,7] or plant communities [8], where deforestation is an extreme change of natural environments.

Most tropical forests are experiencing an increasing rate of deforestation, where the Amazon forest is one of the most threatened [9,10]. Particularly, the Colombian Amazon has been deforested to transform natural forests into cattle pastures [11]. After clear cutting, primary forest is replaced mainly by *Brachiaria decumbens*, an exotic grass species that is highly mycotrophic [12]. Cattle ranching is not sustainable for a long time and, after 5 to

10 years [13], pastures are unproductive, soils are degraded and pastures are abandoned for natural regrowth [14].

Different studies have documented the effect on AM fungal communities during transformation of forests into pastures. Cattle grazing causes plant stress to grass, indirectly affecting AM association. It has been observed that AM fungal sporulation increases in soils with cattle-grazed pastures [15,16] in response to a stressing condition. Based on soil-borne AM fungal spores, Leal et al. [17] additionally found that the composition of AM fungal communities in pastures was different from those in native forest soils, even when the abundance of AM fungal spores in the soil did not change between land-uses. Stürmer and Siqueira [15] found that species richness based on the AM fungal soil-borne spore determination was higher in secondary forests than in primary forests or pastures, and was dominated by two or three fungal species. However, Reyes et al. [18] found small differences in the AM fungal diversity of young, middle-age and mature forests. These results approximated what is occurring in soils, as AM fungal spores reflect an incomplete AM fungal community composition since not all AMF sporulate at the same time and some species do not sporulate at all [19].

Little molecular evidence exists for how AM fungal communities are affected by land-use in the Amazon region. Less drastic land-use changes than deforestation, such as slash-and-burn agriculture indicated that alpha diversity and richness of AM fungal communities were not different from those present in native forests [20]. Additionally, García de León et al. [20] found that pastures had a higher number of generalist AM fungal species than slash-and-burn and native forest soils.

Despite the importance of AMF to terrestrial ecosystems and the important role that these fungi might play in restoration processes, there is little information on how AM fungal communities recover when pastures are abandoned and left for natural regrowth, or how AMF change in soils over the pasture-forest transition. With the aim to understand soil microbial changes in soils of a natural regrowth succession of degraded pastures that were left for natural restoration, we studied changes in the composition, abundance and richness of AM fungal communities. That information is relevant for guiding conservation and restoration processes in the Andean-Amazon region.

## 2. Materials and Methods

### 2.1. Study Area and Sampling Design

The study was conducted in four municipalities of the Caquetá state: Florencia (1°36′50″ N; 75°36′46″ W), Morelia (1°29′09″ N; 75°43′28″ W), Belén de los Andaquíes (1°24′59.1″ N; 75°52′21.2″ W) and San José del Fragua (1°19′52″ N; 75°58′28″ W) (Figure S1; Supplementary Materials), with a total extension of 3000 km$^2$. The study area is characterized by a humid tropical climate, an annual precipitation of 3235 mm, an average temperature of 25 °C and a relative humidity of 80% [21]. The soils are classified as Inceptisols and Oxisols (USDA classification), characterized by a fine texture and drainage limitations. Soils are acidic with a pH between 4.5 and 5.8, with a low base saturation, a high Al saturation and limited contents of C, P, K and Mg [22]. The study area presented two landscapes: (1) Hills with less than 300 m height and slopes between 7 and 12% [22]. This landscape occupied approximately 68% of the Caquetá state. There, the main economic activity is the extensive cattle ranching alternated with agriculture. Anthropogenic landscapes are alternated with relicts of native forest. (2) Mountains higher than 300 m up to 3000 m and slopes between 12 to 75% [22], which are the true Andean-Amazonian transition. The landscape is dominated by native forests, with some cattle pastures, crops and secondary forests [23,24].

Through a random stratified sampling, the optimum number of plots per landscape was obtained: a total of 14 plots in the hills and 19 plots in the mountains. Then, in each landscape unit, a chronosequence was established for cattle pastures with different periods (years) of abandonment: (i) degraded pasture (early natural regrowth stages of degraded pastures or <3-year-old degraded pastures with shrubby vegetation), (ii) 10–20

(succession covered by 10–20-year-old forest or young secondary forest), (iii) 25–40 (succession covered by 25–40-year-old forest or intermediate to mature secondary forest) and (iv) Forest (primary forest). In total, five degraded pastures plots, seven 10–20 succession plots, twelve 25–40 succession plots and nine forest plots were selected. For biodiversity analyses, the categories degraded pasture, 10–20 and 25–40 were pooled and named as "Disturbed", while Forest was named as "Undisturbed". Detailed description of the plots studied appeared in the Table S1 of Supplementary Materials.

Successional stages had different plant composition [25]. The plant composition of degraded pasture plots corresponded to patches of *Brachiaria* spp. grasses with little grazing biomass, intercropped with weeds such as *Urochloa decumbens* (Poaceae), *Homolepsis aturensis* (Poaceae), *Cyperus* sp. (Cyperaceae), *Scleria melaleuca* (Cyperaceae) and *Steinchisma laxum* (Poaceae), and successional secondary shrubs from 34 families such as Melastomataceae (17), Annonaceae (9), Burseraceae (8) and Euphorbiaceae (7). Plant composition included 103 plant species, where the most abundant were: *Miconia elata* (163), *Miconia minutiflora* (156) and *Miconia lourtegiana* (103).

The 10–20 plots were dominated by shrubs and some primary successional tree species. The most abundant plant families were: Melastomataceae (27), Mimosaceae (20), Rubiaceae (18), Moraceae (15), Annonaceae (14), Euphorbiaceae (13), Lauraceae (13), Myrtaceae (10), Flacourtiaceae (9) and Arecaceae, Burseraceae, Caesalpiniaceae, Cecropiaceae, Fabaceae and Fabaceae with 8 species each one. The most abundant plant species were: *Siparuna guianensiss* (138) *Henriettea fascicularis* (89), *Adenocalymma aspericarpum* (76), *Piptocoma discolor* (71) and *Inga thibaudiana* (69).

The 25–40 plots corresponded to a middle-age secondary successional forest, with some Melastomataceae species, but with a high and dense forest. The plant families with higher number of species were Rubiaceae (43), Melastomataceae (40), Mimosaceae (33), Moraceae (33), Fabaceae (30), Annonaceae (28), Lauraceae (27), Burseraceae (18), Clusiaceae (18), Euphorbiaceae (18) and Myristicaceae (17). The dominant plant species were *Tapirira guianensis* (134), *Siparuna guianensiss* (120), *Adenocalymma aspericarpum* (114), *Casearia arborea* (87), *Henriettea fascicularis* (61), *Matayba inelegans* (49) and *Guatteria punctata* (49).

The forest plots corresponded to a primary forest with the highest plant richness, the highest plant height, a well stratified canopy and an understory rich in palms and ferns. The plant families with the higher number of species were: Lauraceae (43), Rubiaceae (39), Melastomataceae (38), Fabaceae (36), Burseraceae (27), Sapotaceae (27), Moraceae (25), Mimosaceae (23), Annonaceae (22), Euphorbiaceae (19), Chrysobalanaceae (16) Elaeocarpaceae (12), Meliaceae (12), Arecaceae (10), Myristicaceae (10), Sapindaceae (9), Caesalpiniaceae (8) Clusiaceae (7), Nyctaginaceae (7) and Lecythidaceae (6). The most abundant plant species were: *Pseudosenefeldera inclinata* (197), *Wettinia praemorsa* (135), *Virola elongata* (87), *Ladenbergia muzonensis* (56), *Graffenrieda colombiana* (50) and *Geonoma maxima* (47).

## 2.2. Soil Sampling

Soil sampling was done in 2018. In each sampling plot, an area of $50 \times 50$ m was delimited. There, a composed soil sample of about 500 g of the topsoil (0–30 cm depth) was collected from three sub-samples for physicochemical analyses. In each sampling plot, five sampling points were defined: four in the corners and one in the center. Individual soil samples of about 500 g of the superficial soil (0–10 cm depth, excluding litter horizon) were collected in each sampling point for the microbial analysis. Soil samples were dried at room temperature and transported in plastic bags to the laboratory, prior to DNA extraction.

## 2.3. Soil Physicochemical Analysis

The following soil variables were evaluated: texture (Bouyoucos); pH (1:1 in water); percentage of organic carbon (OC) (with potassium dicromate solution); percentage of total nitrogen (N total), cation exchange capacity (CEC) in meq/100 g; Ca, Mg, K, N and Na in mg/kg (with ammonium acetate 1N a pH = 7); and available phosphorus (P) in mg/kg (Bray II).

## 2.4. Molecular Analysis

Communitarian soil DNA was extracted from 2 g of soil using a PowerMax® Soil DNA Isolation Kit (Qiagen, Germantown, MD, USA). The DNA was quantified with Nanodrop® (Thermo Fisher Scientific, Madison, WI, USA) before sequencing. AM fungal DNA was amplified from communitarian soil DNA using two specific primers for the small-subunit (SSU ribosomal RNA gene of AMF: WANDA [26] and AML2 [27]. following the methodology described by Davison et al. [6]. PCR products were amplified and checked on 1% agarose gel. PCR products (5 ng/µL) were organized in libraries. Each library was ligated to Illumina adaptors, using the TruSeq DNA PCR-free library prep kit (Illumina Inc., San Diego, CA, USA) and sequenced on the Illumina MiSeq platform, using a $2 \times 300$ bp paired-read sequencing approach. Sequencing was done at the Asper Biogene laboratory (Tartu, Estonia) as described by Peña-Venegas et al. [28].

## 2.5. Bioinformatics

Demultiplexed paired-end reads were analyzed according to the bioinformatics steps described by Vasar et al. [29]. Primer sequences were matched allowing one mismatch in forward or reverse chains after removing barcode and primer sequences, using the PEAR v. 0.9.8. program. Singletons were removed. Amplicons between 170 and 540 pb in length were selected for the analysis. Amplicons longer than 540 bp were cut at that length and included in the analysis. Chimeric sequences were detected and removed using UCHIME v7.0.1090 [30] in the reference database mode using the default parameters and the reference database Maarj*AM* (https://maarjam.botany.ut.ee, accessed on 8 January 2021, status June 2019, 384 virtual taxa (VT)). Sequence alignment was performed using the MAFFT v. 7 multiple sequence alignment web service in JALVIEW version 2.8 [31], subjected to a neighbor-joining phylogenetic analysis in TOPALi v2.5 [32], using the default parameters with two databases: Maarj*AM* and the International Nucleotide Sequence Database Collaboration (INSDC). Retained reads were subjected to a BLAST+ search v 2.8.1 [33], using 97% identity and 95% alignment length thresholds to finally obtain representative sequences of Glomeromycota fungi as virtual taxa (VT).

## 2.6. Statistical Analysis

The sampling effort was evaluated for the successional categories and landscapes using accumulative species curves and the Exact (Mao Tao) method (accumcomp function from package Biodiversity R [34] in R 4.0.3 language [35]). The AM fungal community composition and diversity in the different landscapes and successional categories were described by AM fungal richness and exponential Shannon-Weiner, inverse Simpson and Pielou evenness indexes, using the *diversity* function from R package vegan [36]. Venn diagrams were performed with the *ggvenn* function from R package ggven [37] and barplots were performed in InfoStat v.2020 software [38], to visualize the richness in each successional category, landscape and ecosystem. Linear Mixed-Effects (LME) models were adjusted to compare diversity indexes with soil physicochemical properties, using plots as aleatory effects. Normality and homeostacity were validated using QQ-plot and fitted-plot exploratory analyzes of residuals. Media were separated using a LSD Fisher test with 5% of significance in InfoStat v.2020. Models obtained were adjusted with the *lme* function from R package nlme [39] using the interface in InfoStat v.2020. Differences in the AM fungal communities between successional stages and landscapes were analyzed using nested PERMANOVA tests with 999 permutations and the *nested.npmanova* function from the R package Biodiversity R. A non-multidimensional analysis (based on Bray-Curtis distance and 50 iterations) was used to explore the variability of the AM fungal community composition, using the functions *metaMDS* and *ordiellipse* from R package vegan. Indicator species values for AM fungal VT were obtained for landscapes and successional stages, using the *indval* function from R package labdsv [40]. Canonical correlation analysis (CCA), Mantel's test (Method: Pearson correlation and 9999 permutations) and Pearson correlation analysis were performed to analyze associations between ecological variables and soil properties.

Mantel's test was performed on the Bray-Curtis and Euclidean distances with the *mantel* function from R package vegan. The CCA and Pearson correlation were performed in InfoStat v.2020. Visualization of the Pearson correlation matrix was done with corrplot [41] and circlize packages [42] in R language. Finally, redundancy analyzes evaluated the effect of soil properties on the AM fungal communities (Hellinger transformation and 999 permutations), with gridplot graphics using *rda*, *ordiellipse*, *envfit* and *ordisurfgrid.long* functions from R package vegan.

## 3. Results

### 3.1. Soil Conditions

Soils of the study area were very acidic (pH between 3.84 and 4.26), with limited organic carbon (<3%) and limited amounts of macroelements (N, P, K, Ca and Mg). Total nitrogen (<0.20%) and available phosphorus (<6.1 mg/kg) were the limited nutrients in these soils. Then, soils were classified as acidic nutrient-limited soils. Landscape and chronosequence interaction was not significant for any soil variable ($p > 0.05$) (Table 1). Hill and mountain soils were different in their texture and CEC. Soils from natural regrowth stages were different in pH, OC, total N, K and percentage of loam from soils of degraded pastures.

**Table 1.** Soil physico-chemical composition of two different landscapes and four different stages of natural regrowth of degraded pastures.

| Factor | Level | pH | CEC (meq/100 g) | OC (%) | Total N (%) | Ca (mg/kg) | Mg (mg/kg) |
|---|---|---|---|---|---|---|---|
| Landscape | Hill | 4.22 ± 0.11 a | 7.35 ± 0.55 a | 2.29 ± 0.20 a | 0.19 ± 0.02 a | 320.75 ± 51.62 a | 45.17 ± 20.11 a |
| | Mountain | 4.04 ± 0.10 a | 5.55 ± 0.46 b | 2.04 ± 0.17 a | 0.17 ± 0.01 a | 236.42 ± 59.78 a | 76.54 ± 21.87 a |
| Chronosequence | Degraded pasture | 4.26 ± 0.10 a | 4.97 ± 0.49 a | 1.67 ± 0.20 b | 0.14 ± 0.02 b | 236.92 ± 24.79 a | 39.67 ± 3.46 a |
| | 10–20 | 4.25 ± 0.26 ab | 7.38 ± 1.06 a | 2.19 ± 0.36 a | 0.18 ± 0.03 ab | 394.83 ± 154.03 a | 106.25 ± 66.24 a |
| | 25–40 | 4.15 ± 0.08 ab | 7.27 ± 0.49 a | 2.37 ± 0.13 a | 0.20 ± 0.01 a | 253.00 ± 15.68 a | 52.25 ± 5.05 a |
| | Forest | 3.84 ± 0.05 b | 6.19 ± 0.67 a | 2.42 ± 0.29 a | 0.20 ± 0.02 a | 229.58 ± 19.20 a | 45.25 ± 4.96 a |

| Factor | Level | Na (mg/kg) | P (mg/kg) | K (mg/kg) | Texture | | |
|---|---|---|---|---|---|---|---|
| | | | | | Loam (%) | Clay (%) | Sand (%) |
| Landscape | Hill | 43.38 ± 2.64 a | 5.69 ± 0.53 a | 58.18 ± 6.55 a | 14.21 ± 2.62 a | 44.29 ± 3.47 a | 41.50 ± 4.18 b |
| | Mountain | 45.04 ± 2.04 a | 5.32 ± 0.43 a | 71.21 ± 5.58 a | 13.67 ± 2.19 a | 21.50 ± 2.84 b | 64.83 ± 3.50 a |
| Chronosequence | Degraded pasture | 45.25 ± 2.64 a | 4.86 ± 0.74 a | 47.76 ± 4.44 b | 7.75 ± 0.77 b | 28.75 ± 6.99 a | 63.50 ± 6.64 a |
| | 10–20 | 44.58 ± 1.72 a | 6.01 ± 0.76 a | 75.13 ± 10.96 a | 16.17 ± 4.53 ab | 32.50 ± 2.96 a | 51.33 ± 5.86 a |
| | 25–40 | 44.17 ± 2.51 a | 5.06 ± 0.42 a | 72.49 ± 8.89 a | 16.33 ± 3.08 a | 35.00 ± 2.83 a | 48.67 ± 4.42 a |
| | Forest | 45.83 ± 5.33 a | 6.09 ± 0.76 a | 63.38 ± 8.82 ab | 15.50 ± 4.00 ab | 35.33 ± 3.84 a | 49.17 ± 4.59 a |

CEC: cation exchange capacity, OC: organic carbon, Total N: total nitrogen, Ca: calcium, Mg: magnesium, Na: sodium, P: available phosphorus, K: potassium. Values in columns corresponded to mean and standard error. Values followed by the same letter do not differ statistically (Fisher's least significant difference LSD test, $p < 0.05$).

### 3.2. AM Fungal Sequencing Data

A total of 59,199 reads were obtained from the 65 soil samples analyzed. Reads corresponded to 64 AM fungal virtual taxa (VT) of seven families, ten genera and ten species. The richest genus was *Glomus* (45 VT), followed by *Acaulospora* (6 VT) and *Paraglomus* (3 VT). *Archaeospora*, *Rhizophagus* and *Scutellospora* were represented in the fungal community with two VT each one and *Ambispora*, *Claroideoglomus*, *Gigaspora* and *Kuklospora* only with one VT, respectively (Table S2, Supplementary Materials). *Paraglomus* sp. was the most abundant genus with 37,386 reads, representing the 63% of the total arbuscular mycorrhizal fungal community, followed by *Glomus* VT223 with 8188 reads.

### 3.3. AM Fungal Abundance, Richness and Diversity

According to the species accumulation curves, the sampling effort was enough to properly describe the composition and diversity of AM fungal communities in the different categories, except for degraded pastures and 10–20 plots, as in those two categories less soil samples were collected (Figure S2, Supplementary Materials).

The 73% of the AM fungal community was composed by Glomeraceae VT. Glomeraceae dominated most of the AM fungal community composition of the soils evaluated. In undisturbed ecosystems, the genera *Ambispora*, *Claroideoglomus* and *Kuklospora* were

absent (Figure 1a). The genera *Scutellospora* and *Kuklospora* were present in the mountain soils but not in the hill soils (Figure 1b). The genus *Ambispora* was only recuperated in hill soils. Soils of degraded pastures included all the AM fungal genera recovered in this study (Figure 1c). During the chronosequence, some genera disappeared, and others reappeared over time.

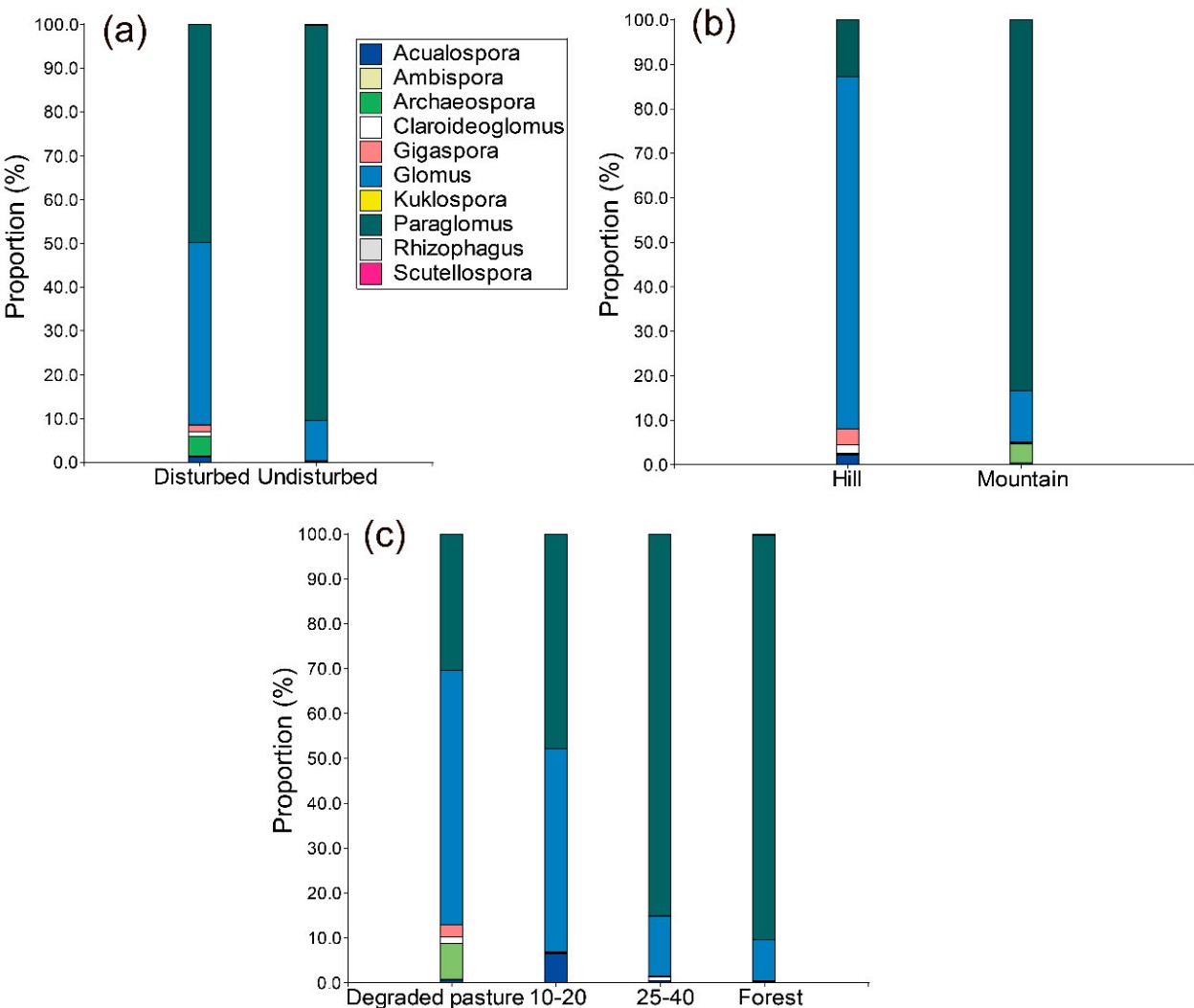

**Figure 1.** Proportion (%) of the abundance of reads of arbuscular mycorrhizal (AM) fungal genera in the AM fungal community composition of different levels of ecosystem disturbance (**a**); landscapes (**b**); and among successional categories of natural regrowth (**c**).

During the natural regrowth succession, a decrease of *Glomus* representativeness in the soil occurred (Figure 1c): in degraded pastures *Glomus* represented 56.6% of the AM fungal community, in 10–20 it represented the 45.4%, in 25–40 it represented the 13.4% and in forest soils it represented only the 9.3%. Additionally, *Paraglomus* increases its representativeness in the soil over the natural regrowth chronosequence (Figure 1c). In soils of degraded pastures *Paraglomus* represented the 30.3% of the AM fungal community, in 10–20 it represented the 47.7%, in 25–40 it represented 85% and in forest soils it represented the 90.2%.

Soils from disturbed ecosystems held 50% more VT than undisturbed ecosystems. Mountain soils harbored more VT (42,466 reads and 56 VT) than hill soils (16,733 reads and 41 VT). Degraded pastures had more abundance of VT in their soils (22,498 reads), than successional categories (10–20 with 4356 reads and 25–40 with 12,576 reads) or mature forest soils (19,769 reads). The 10–20 category had more VT in the soils (a total of 42 VT)

than degraded pastures (38 VT), 25–40 successional forests (37 VT) or mature forests (29 VT). VT richness was not related with VT abundance.

AM fungal community composition of landscapes was different. Mountain soils were richer in VT than the hill soils. The 10–20 category was the natural regrowth stage where soils presented more exclusive VT and where the maximum number of VT in the soil was reported (Figure 2).

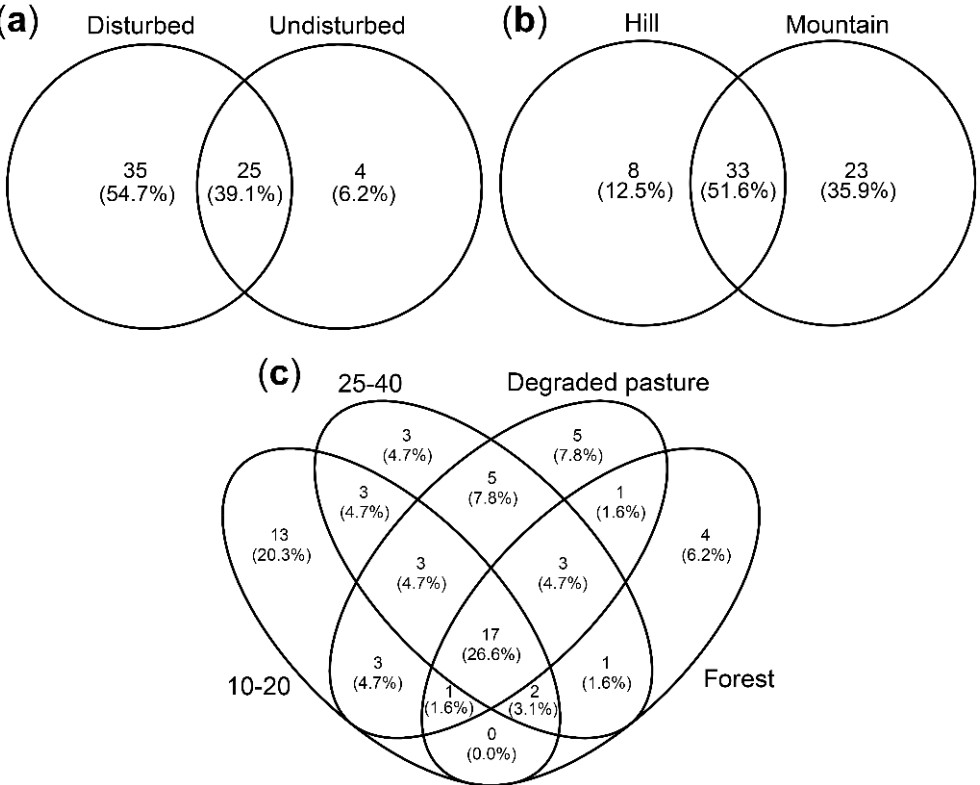

**Figure 2.** Venn diagrams for the VT composition of arbuscular mycorrhizal fungal community in: (**a**) disturbed vs. undisturbed ecosystems; (**b**) hill vs. mountain landscapes; (**c**) natural regrowth successional stages (degraded pastures, 10–20 years, 25–40 years, forest) of degraded pastures.

Landscape and chronosequence interaction was not significant for any diversity index ($p > 0.05$) (Table 2). Degraded pasture soils had a significantly higher VT richness than soils from less disturbed regrowth stages. Diversity indexes were similar during all stages of the chronosequence.

**Table 2.** Arbuscular mycorrhizal fungal richness and diversity indexes according to the type of landscape and successional stages of degraded pastures natural regrowth.

| Factor | Level | Richness | Exponential Shannon (expH′) | Inverse Simpson 1/D | Equitability (Evenness-Piélou) |
|---|---|---|---|---|---|
| Landscape | Hill | 7.70 ± 1.04 a | 3.15 ± 0.573 a | 2.55 ± 0.41 a | 0.49 ± 0.07 a |
| | Mountain | 6.50 ± 0.96 a | 2.45 ± 0.46 a | 1.94 ± 0.33 a | 0.32 ± 0.06 a |
| Chronosequence | Degraded pasture | 12.12 ± 2.07 a | 3.43 ± 1.06 a | 2.50 ± 0.69 a | 0.53 ± 0.09 a |
| | 10–20 | 6.60 ± 1.22 b | 3.57 ± 0.73 a | 2.89 ± 0.58 a | 0.46 ± 0.07 a |
| | 25–40 | 4.83 ± 0.52 b | 2.33 ± 0.37 a | 2.05 ± 0.29 a | 0.34 ± 0.12 a |
| | Forest | 4.83 ± 1.04 b | 1.88 ± 0.62 a | 1.54 ± 0.46 a | 0.29 ± 0.09 a |

Values in the columns corresponded to mean and standard error. Values followed by the same letter do not differ statistically (Fisher's LSD test, $p < 0.05$).

### 3.4. AM Fungal Community Composition

The PERMANOVA within landscapes indicated that there were no significant differences in the AM fungal composition of natural regrowth successional stages in hills ($p = 0.468$) or mountains ($p = 0.209$), as it is shown in the NMDS plots (Figure S3, Supplementary Materials). Therefore, successional stages of the two landscapes were evaluated together.

The PERMANOVA corroborated that AM fungal communities of the two landscapes were significantly different ($p < 0.01$) (Figure 3a), as it is shown in the NMDS plot (S = 0.15), but not among natural regrowth successional stages ($p > 0.05$). Additionally, the NMDS plot indicated a high similarity of AM fungal communities of natural regrowth stages, but not with the AM fungal community of degraded pastures (Figure 3b).

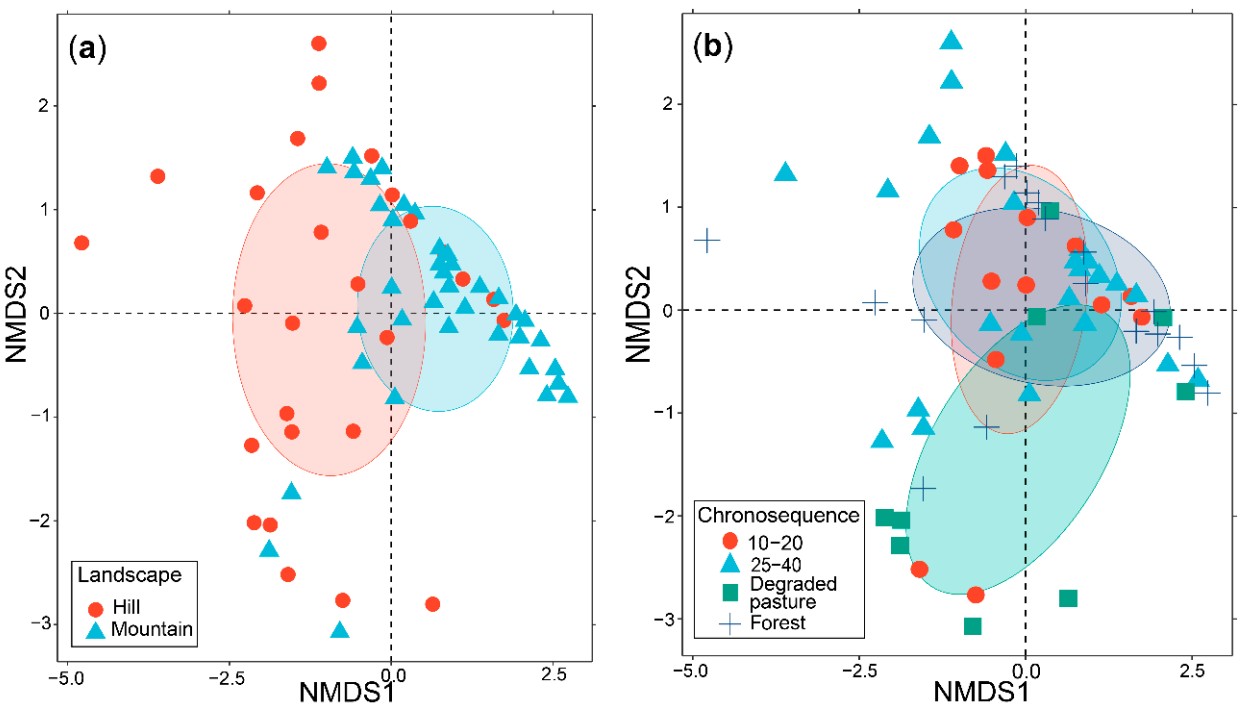

**Figure 3.** Non-metric multidimensional scaling (NMDS) plots displaying the composition of AM fungal communities in different landscapes (**a**) and different successional stages of natural regrowth of degraded pastures (**b**). Ellipses indicated the standard deviation around the centroid of each variable.

The indicator species value found two VT of AMF characteristic of the mountain landscape (*Paraglomus* VT444) and one characteristic of the hill landscape (*Glomus* VT70). Additionally, four *Glomus* and one *Archaeospora* VT were indicator species of degraded pastures (Table 3).

**Table 3.** Indicator species of AM fungal VT in landscapes and natural regrowth stages of degraded pastures.

| Factor | Level | Virtual Taxon | Identity | Indicator Value | Probability |
|---|---|---|---|---|---|
| Landscape | Mountain | VT444 | *Paraglomus* sp. | 0.9225 | 0.002 |
| | Hill | VT70 | *Glomus* sp. | 0.2480 | 0.029 |
| Chronosecuence | Degraded pasture | VT89 | *Glomus* sp. | 0.4239 | 0.011 |
| | | VT126 | *Glomus* sp. | 0.4095 | 0.030 |
| | | VT4 | *Archaeospora* sp. | 0.3326 | 0.010 |
| | | LH.Cl01 | *Glomus* sp. | 0.2914 | 0.018 |
| | | VT93 | *Glomus* sp. | 0.2556 | 0.014 |

Virtual taxa with an indicator value >0.25 were considered as indicator species.

### 3.5. Relationship between Soil Properties and AM Fungal Communities

The canonical correlation analysis showed an important association between soil physicochemical parameters and diversity indices ($r = 0.75$, $p < 0.05$). The Mantel test showed a significant correlation between AM fungal community composition (Bray-Curtis distance) and the percentage of clay in soils (Euclidean distance) ($R = 0.13$; $p < 0.05$). Therefore, the amount of clay in the different soils was a soil feature differentiating AM fungal communities (Figure S4, Supplementary Materials).

From all the soil properties evaluated, pH was the only variable that was positively correlated with ExpShannon, invSimpson (1/D) and evenness indexes (Figure 4). Additionally, Mg and Ca were variables related with ExpShannon and invSimpson indexes.

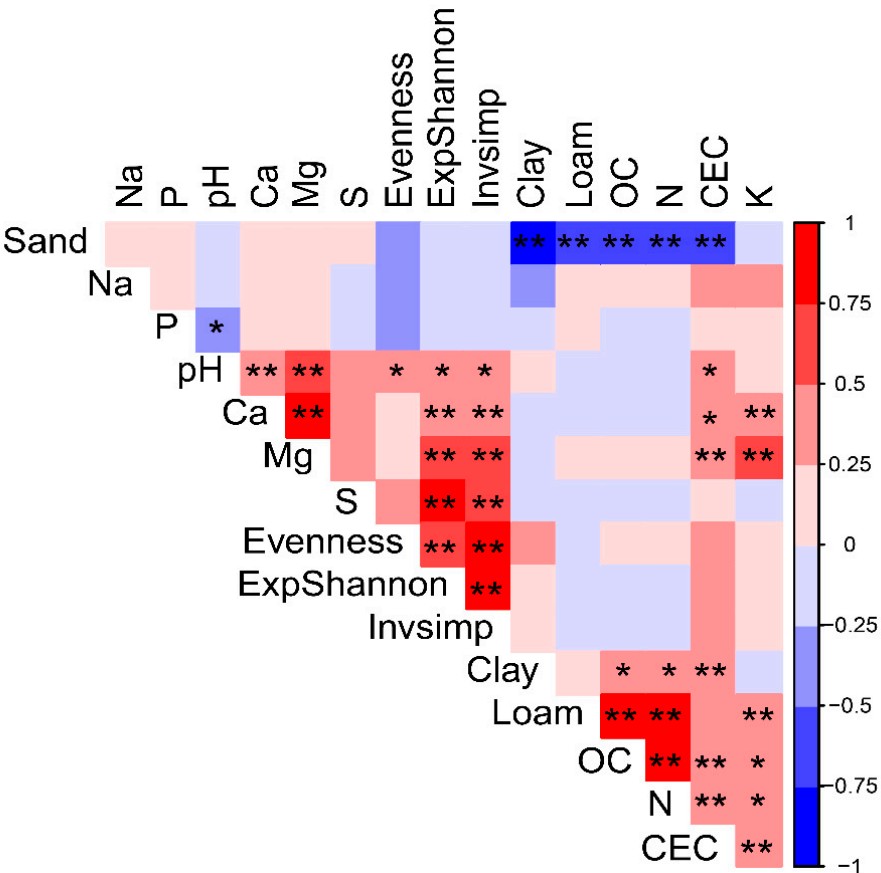

**Figure 4.** Pearson correlation correlogram of the AM fungal diversity indexes and soil physicochemical variables (Red colors: positive interactions, blue colors: negative interactions). Asterix represents the significance of the correlation * significant, $p < 0.05$; ** high significant, $p < 0.01$.

The redundancy analysis (RDA) of AM fungal communities and soil physicochemical properties (constrained inertia = 37.6%) clearly showed the differences for the AM fungal composition of the two landscapes evaluated (Figure 5a), but not for the natural regrowth successional stages (Figure 5b). *Paraglomus* VT444 was highly related to the mountain landscape with sandier soils. *Glomus* VT89 and VT126 were highly related with the hill landscape. Although *Glomus* VT359 was recovered in all the landscapes and all the stages of natural regrowth, it was more related to old stages of the natural regrowth succession where more OC and N and a better soil texture (loam) occurred.

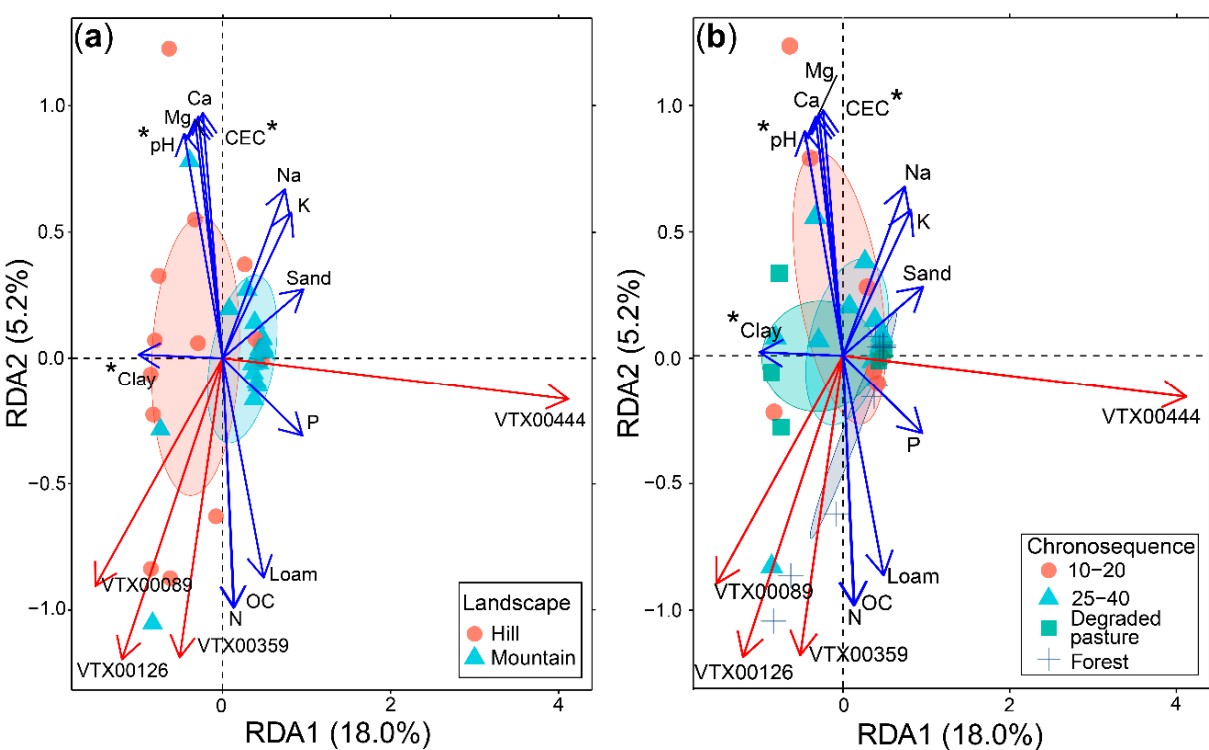

**Figure 5.** Redundancy analysis (RDA) of arbuscular mycorrhizal fungal communities in relation to landscapes (**a**) and successional natural regrowth stages (**b**), constrained by soil variables. Blue arrows correspond to significant soil variables for the constrained ordination ($p < 0.05$). Red arrows correspond to significant AM fungal VT species, with a maximum correlation with ordination scores ($r \geq 0.5$; $p < 0.05$). Ellipsoids indicate standard deviation around the centroid of each variable. Asterixs represent significant differences in those variables for the RDA analysis.

A gradient analysis of significant soil variables based on RDA results (Figure S5, Supplementary Materials), corroborated the differences in the soil texture of the two landscapes. Results also explain how plots located in one of these two landscapes were influenced by their soil composition. As more degraded pastures were collected in the hills, plots of the chronosequence collected there were highly influenced by clayed soils of hills. In the same way, more soil samples of natural regrowth stages were collected in the mountain landscape and therefore, plots of the chronosequence collected there were highly influenced by sandier soils of mountain.

## 4. Discussion

Despite the advantages from the use of molecular approaches to study the AM fungal communities in soils of a successional natural regrowth chronosequence, our results corroborate previous results obtained from soil-borne AM fungal spores. Hills and mountains are continuous landscapes in the Andean-Amazonian transition with different soil texture. Soil texture was an important soil feature for AM fungal community differentiation where AM fungal communities differed in composition and VT richness. Although changes in AM fungal diversity were similar over the different stages of the natural regrowth chronosequence, differences in the representativeness of the genera *Glomus* and *Paraglomus* were evidenced.

In recent years, the use of molecular methodologies to study non-cultivable microbial communities has becoming more popular. This is the case of AM fungal communities which traditionally had been studied indirectly from soil-borne spores. Isolating AM fungal spores is the easiest way to access to AM fungal communities present in the soil, but the spores did not offer the complete composition of AM fungal communities [43,44]. Using molecular tools instead of spore-dependent tools, relevant and new information on AM fungal community composition in natural regrowth successional stages was expected.

However, results corroborated that the degraded pastures had the highest abundance of AMF in the soil matrix (50% more VT than soils from successional regrowth stages), even with *Brachiaria* grasses as a dominant cover, which is a highly mycotrophic species [12]. This result might be reflecting the stressful environmental conditions for the AM symbiosis (low soil acidity and low available P with less than 5 mg/kg). The endophytic nature of fungi would obligate them to continue colonizing plant roots but at the same time, searching for better conditions and moving into the soil to access new plant hosts.

Contrary to what was reported by Leal et al. [17], there were differences in the abundance of AMF among land uses. Differences might be the result of the approach used to assess AM fungal abundance. AM fungal communities in the study area were dominated by Glomeraceae species, which produce high numbers of spores, so most spores recuperated must came from those species. When molecular approaches are used, this bias does not exist, reflecting the true abundance of AMF in the soil evaluated. Our results showed that less abundance, but high richness of AMF occur in early successional stages of the natural regrowth succession. In early successional stages (10–20), when the plant community began to diversify, more AM fungal species start to colonize new host plants aside from grasses, increasing VT richness into the soil matrix but decreasing VT abundance in it as fungi will move preferentially to plant roots. As previously reported [15], the secondary forests presented a higher AM fungal richness than primary forests. These results might be related with the plant composition of secondary forest (young and pioneer plant species) which is more active metabolically [45,46] and therefore, more attractive to AM fungi as host plant. In mature forests, the plant community is dominated by old plant species with well-consolidated AM symbiosis in their roots and low soil disturbance. Therefore, AMF in mature forests will be principally into plant roots rather than in the soil matrix.

It was also found that AM fungal communities of degraded pastures were different from those of natural regrowth successional stages, even when trees were not dominant in the plant community. Among all the soil variables evaluated, pH was the one significantly different between degraded pastures and successional regrowth stages. Les acidic soils were found in degraded pastures than in successional regrowth stages. Soil pH is one of the main abiotic drivers differentiating AM fungal communities [6,47]. Contrary to other natural environments, in Amazonian soils, soil acidification is a signal of soil restoration and therefore, a natural soil condition for native AM fungal communities. Disturbed ecosystems (as degraded pastures and successional regrowth stages) and undisturbed ecosystems (as forests) only share 39% of the AM fungal community (Figure 2). It has been indicated that pastures use to hold more generalist AMF than more conserved environments [20]. In our study, *Ambispora leptoticha* was one of the few VT recovered only in degraded pastures. *A. leptoticha* is common in tropical environments [14,48], and frequently recovered in Amazonian soils [49,50] of disturbed areas such as pastures, crops, recovering mining areas, and young secondary forests [15,50,51]. Therefore, it could be considered a generalist AM fungal species in tropical soils.

Although there were differences in the AM fungal communities in the different natural regrowth successional stages, the main differences among AM fungal communities were the result of differences in the soil composition (texture) of the landscapes sampled (Figure 5). Soil texture is one of the least features studied in relation to AM fungal communities. Differences in soil texture will affect soil atmospheric pressure, moisture and therefore, the nutrient availability and nutrient mobility into the soil, affecting microbial communities [52]. It has been reported that sandier soils offer better conditions for AMF because of better soil aeration and pore size, which are reflected in a higher percentage of root colonization [53]. However, Okiobé et al. [54] indicated that clayed soils favor AM fungal sporulation. Our results cannot provide information on root colonization or AM fungal sporulation, but clearly identified AMF that are favored by one of the landscape conditions. The mountain harbored 2.5 more AM fungal reads than the hills, and 15 more VT (26% of the total VT obtained). These results could indicate that sandy soils are better soil environments for soil-borne AM fungal communities than clay soils. However, floristic composition of the

landscapes might also influence the abundance and richness of AMF present in soils. It was also found that *Paraglomus* was the most abundant genus (with 37,386 reads) in the mountain landscape, and *Paragomus* VT444 a species indicator of it. *Glomus* was most abundant in the hill landscape with clayey soils. The genera *Scutellospora* and *Kuklospora* were only present in mountain soils. These two genera produce few big spores that, because of their size, will be positively affected by bigger soil pores and sandier soils.

Although the AM fungal communities in hills and mountains were different, they changed similarly during the natural regrowth succession in the two landscapes. *Glomus* was usually the dominant genus in the soil-borne AM fungal community during the young successional stages of the chronosequence, but its presence in the soil decreased over time (from 56.6% in degraded pastures to 9.3% in forest soils), and other AM fungal genus such as *Paraglomus* increases its representativeness in the AM fungal community (from 30.3% in degraded pastures to 90.2% in forest soils). *Glomus* has been reported as one of the dominant genera in successional forest stages in the Amazon region [15,18,51]. *Glomus* is a very diverse genus with proliferous spore-producer species. However, it is not easy to distinguish taxonomically spore morphotypes within *Glomus* and among other genera that produce spores with hyphal connections. Molecular tools made possible to distinguish clearly *Glomus* from other spore-producer genera with similar spore morphologies such as *Rhizophagus* and *Paraglomus* [19,55]. These advances, allowed in this study to discriminate properly *Glomus* from *Paraglomus* sequences, and showed changes in the representativeness of each genera into the AM fungal community. A reduction of *Glomus* abundance in the soil during the successional regrowth might reflect less stressing environmental conditions that reduced the need of *Glomus* to sporulate [16]. *Paraglomus* has been associated with forested areas [14,20,28] and its increasing abundance in the soil during the successional regrowth could reflect the preference of this genus for more diverse and forested environments. These two particular changes in the AM fungal community composition over the successional chronosequence might be considered signals of environmental restoration.

## 5. Conclusions

Molecular and spore-dependent approaches to study soil-borne AM fungal communities are accurate and provide similar information on changes that AM fungal communities experienced during the natural regrowth of degraded pastures. As expected, in degraded pastures with stressing soil conditions, a higher abundance of AMF occurred into the soil than in soils of successional regrowth stages. Soils of successional regrowth stages presented a higher AM fungal VT richness, in concordance with a more metabolically active and diverse plant community. Although there were differences in soil composition and AM fungal community composition of the landscapes evaluated, changes in the AM fungal communities during the chronosequence were similar. Distinctive abiotic and biotic characteristics indicate soil restoration over the time such as the soil acidification, the reduction of *Glomus* dominance, and a higher representativeness of *Paraglomus* in the soil-borne AM fungal community.

**Supplementary Materials:** The following are available online at https://www.mdpi.com/article/10.3390/agronomy11122468/s1, Figure S1: Map of the study area (Caquetá state, Northwestern Colombian Amazon); Figure S2: Accumulation curves (Mau-Tau method with a 95% confidence interval) of arbuscular mycorrhizal fungal species in soils of different landscapes (a), and different successional stages of a degraded pasture natural regrowth (b); Figure S3: Non-metric multidimensional scaling (NMDS) plots displaying the composition of arbuscular mycorrhizal fungal communities in the hill landscape (a) and the mountain landscape (b) over successional stages of a degraded pasture natural regrowth; Figure S4: Mantel correlation scatter-plot between the dissimilarity of arbuscular mycorrhizal fungal VT species abundance (Bray-Curtis distance) and dissimilarity of the percentage of clay (Euclidean distance); Figure S5: Redundancy analysis (RDA) of arbuscular mycorrhizal fungal species according to a soil (pH, clay and CEC) gradient in the different successional stages of a degraded pasture natural regrowth (a–c), and landscapes (d–f); Table S1: Soil samples collected and analyzed to study arbuscular mycorrhizal fungal communities; Table S2: Arbuscular mycorrhizal

fungal virtual taxon (VT) recovered from two landscapes and four successional stages in a natural regrowth chronosequence of abandoned pastures in the Andean-Amazonian transition. Abundance as the number of read VT accounts.

**Author Contributions:** Conceptualization, C.H.R.-L. and C.P.P.-V.; methodology, C.P.P.-V.; software, A.S.; validation, A.S. and C.P.P.-V.; formal analysis, A.S.; investigation, C.H.R.-L., C.P.P.-V., H.M.-R. and Y.R.V.-D.; resources, C.H.R.-L.; data curation, C.P.P.-V., H.M.-R., Y.R.V.-D. and A.S.; writing—original draft preparation, C.H.R.-L., C.P.P.-V. and H.M.-R.; writing—review and editing, C.H.R.-L., C.P.P.-V. and A.S.; visualization, A.S. and H.M.-R.; supervision, C.H.R.-L. and C.P.P.-V.; project administration, C.H.R.-L.; funding acquisition, C.H.R.-L. All authors have read and agreed to the published version of the manuscript.

**Funding:** This research was part of the project: "Restauración de áreas disturbadas por implementación de sistemas productivos agropecuarios en zonas de alta intervención en el Caquetá", funded by the Fondo de Ciencia, Tecnología e Innovación FCTeI—SGR, the Amazonian Scientific Research Institute Sinchi, the Government of Caquetá, the Universidad de la Amazonía, the Asociación de Reforestadores y Cultivadores de Caucho del Caquetá ASOHECA, and the Federación Departamental de Ganaderos del Caquetá FEDEGANGA. Contract 60/2013; and by the Government of Colombia through the project BPIN 2017011000137 "Investigación en conservación y aprovechamiento sostenible de la diversidad biológica, socioeconómica y cultural de la Amazonia colombiana".

**Data Availability Statement:** Data are available from the authors upon request.

**Acknowledgments:** The authors thank all the farmers of the study area for their help and support during the fieldwork and Christopher King for reviewing the English version of this manuscript.

**Conflicts of Interest:** The authors declare no conflict of interest. The funders had no role in the design of the study; in the collection, analyses, or interpretation of the data; in the writing of the manuscript, or in the decision to publish the results.

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
