# Peer review of "Changes in Soil-Borne Communities of Arbuscular Mycorrhizal Fungi during Natural Regrowth of Abandoned Cattle Pastures Are Indicative of Ecosystem Restoration"

_agronomy, doi:10.3390/agronomy11122468_

Round 1

Reviewer 1 Report

The manuscript entitled ‘Changes in soil-borne communities of arbuscular mycorrhizal fungi during natural regrowth of abandoned cattle pastures are indicative of ecosystem restoration’ fits within the general scope of Agronomy MDPI. The article evaluated the soil microbial changes in soils of a natural regrowth succession of degraded pastures that were left for natural restoration. The authors, focused the attention on the changes in the composition, abundance and richness of AM fungal communities.

The manuscript is high originality and the work was well conducted. In the current experiment a high quality analytical methods were used and the experimental design and statistical analysis sounds; the manuscript is of high significance to the field and high interest to general audience, and the quality of writing is acceptable.

I have minor suggestions and comments on the paper .

  • The article has some formatting issues. Please check it;
  • In order to reduce the length of the manuscript, please move some Figure 1 in the supplementary file;
  • Line 378: please rewrite the follow sentence using other words: “In recent years, there has been a boom…”

Based on the above considerations, the paper is well written, the analytical methods are of high quality and the paper could be considered in Agronomy after minor revisions.

Reviewer 2 Report

L22: “Molecular techniques”. Which techniques? Please specify.

L25: “More AMF”. Does this means more AMF species? Please clarify.

L139: “y”?
